# Advances in Early Detection of Pancreatic Cancer

**DOI:** 10.3390/diagnostics9010018

**Published:** 2019-02-05

**Authors:** Atsushi Kanno, Atsushi Masamune, Keiji Hanada, Masataka Kikuyama, Masayuki Kitano

**Affiliations:** 1Division of Gastroenterology, Tohoku University Graduate School of Medicine, 1-1 Seiryo-machi, Aoba-ku, Sendai 980-8574, Japan; amasamune@med.tohoku.ac.jp; 2Japan Study Group on the Early Detection of Pancreatic Cancer (JEDPAC), Onomichi 722-8508, Japan; kh-ajpbd@nifty.com (K.H.); kikuyama110@yahoo.co.jp (M.K.); kitano@wakayama-med.ac.jp (M.K.); 3Department of Gastroenterology, JA Onomichi General Hospital, Onomichi 722-8508, Japan; 4Department of Gastroenterology, Tokyo Metropolitan Cancer and Infectious Diseases Komagome Hospital, Tokyo 113-0021, Japan; 5Department of Gastroenterology, Wakayama Medical University School of Medicine, Wakayama 641-0012, Japan

**Keywords:** pancreatic ductal cell carcinoma, PDAC, early stage, ERCP

## Abstract

Pancreatic ductal adenocarcinoma (PDAC) is a lethal disease. PDAC is the fourth leading cause of death in the United States and Japan based on epidemiological data. Early detection of PDAC is very important to improve the prognosis of PDAC. Early detection of pancreatic ductal adenocarcinoma (PDAC) requires further examination after selecting cases with risk factors for the condition, such as family history, hereditary pancreatic carcinoma syndrome, intraductal papillary mucinous neoplasms, or chronic pancreatitis. The Japan Study Group on the Early Detection of Pancreatic Cancer has investigated and clarified the clinicopathological features for the early diagnosis of PDAC. In Japan, an algorithm for the early diagnosis of PDAC, which utilized the cooperation of local clinics and regional general hospitals, has been a breakthrough in the detection of early-stage PDAC. Further approaches for the early diagnosis of PDAC are warranted.

## 1. Introduction

Pancreatic ductal adenocarcinoma (PDAC) has an extremely poor prognosis, and the associated death toll continues to increase despite developments in treatment methods and diagnostic modalities [1,2]. This is primarily due to the difficulty of diagnosing PDAC in its early stages. However, several recent reports on early-stage PDAC have been published, and an improvement in prognosis for PDAC is expected. Here, we present several approaches to diagnosing PDAC in its early stages.

## 2. Epidemiology

According to cancer statistics in the United States [1], the overall number of patients with cancer is still increasing. The number of deaths caused by PDAC is gradually increasing, affecting approximately 44,330 individuals per year. This makes PDAC the fourth leading cause of death in the United States. Based on global data in 2018 (GLOBOCON), PDAC is the fourteenth most common cancer type, with 458,918 new cases, and the seventh most frequent cause of cancer-related mortality with 432,242 deaths [2]. Moreover, the age-standardized rate (ASR) incidence and ASR mortality associated with PDAC are higher in Japan than in other countries worldwide [3,4], as shown in Table 1.

On the other hand, original Japanese cancer statistical data have been estimated with accuracy by data collected from each prefecture [5]. According to these data [5], in 2016, malignant neoplasms of all organs caused 372,986 deaths, highlighting a constant increase [1]. The overall cancer incidence in 2014 was as follows: (1) colorectal cancer, 135,434 cases; (2) gastric cancer, 129,239 cases; (3) lung cancer, 114,550 cases; (4) breast cancer, 78,529 cases; (5) prostate cancer, 74,459 cases; (6) liver cancer, 40,666 cases; (7) pancreatic carcinoma, 36,239 cases; (8) lymphoma, 28,486 cases; (9) uterine carcinoma, 25,784 cases; (10) esophageal carcinoma, 22,784 cases; (11) gall bladder or bile duct carcinoma, 22,257 cases; (12) urinary bladder carcinoma, 20,107 cases; (13) thyroid cancer, 15,816 cases; (14) leukemia, 12,068 cases; (15) ovarian carcinoma, 10,048 cases; and (16) others, 110,247 cases. Thus, PDAC has the seventh-highest rate of incidence, as shown in Figure 1 [5]. The overall number of cancer-related deaths in 2016 was as follows: (1) lung cancer, 73,838 cases; (2) colorectal cancer, 50,099 cases; (3) gastric cancer, 45,531 cases; (4) pancreatic carcinoma, 33,475 cases; (5) liver carcinoma, 28,528 cases; (6) gall bladder or bile duct carcinoma, 17,965 cases; (7) breast cancer, 14,015 cases; (8) lymphoma, 12,384 cases; (9) prostatic cancer, 11,803 cases; (10) esophageal carcinoma, 11,483 cases; (11) leukemia, 8801 cases; (12) uterine carcinoma, 6345 cases; (13) ovarian carcinoma, 4758 cases; (14) thyroid carcinoma, 1779 cases; and (15) others, 52,182 cases. Thus, PDAC occupies fourth place, as shown in Figure 2. Moreover, PDAC accounts for 8.6% of all deaths [6], with sex-related mortality at 28.0/10,000 individuals for men and 25.6/10,000 individuals for women. Although the prognosis of malignant tumors in other organs has improved owing to the development of various treatment methods, the prognosis of PDAC remains extremely poor. Moreover, while the 10-year relative survival rates per organ published by the National Cancer Center Hospital were 69.8% for colorectal cancer, 69.0% for gastric cancer, and 80.4% for breast cancer, the rate for PDAC was only 4.9% [5]. Furthermore, the overall 5-year survival rate for resected patients with PDAC from 2001 to 2007 was 18.8% and the 5-year survival rate for non-resection cases was extremely low at 3.1% [7].

Meanwhile, according to the Japanese pancreatic cancer registry of the Japan Pancreas Society, the 5-year survival rates for Union for International Cancer Control (UICC) stages 0, Ia, and Ib are 85.8%, 68.7%, and 59.7%, respectively, which are relatively favorable compared with the overall prognosis of PDAC. This demonstrates that early diagnosis and resection can help improve prognosis [7]. However, UICC stage 0, Ia, and Ib cases of all PDAC account for just 1.7%, 4.1%, and 6.3%, respectively [7]. These data revealed the difficulty in diagnosing PDAC in its early stages. The early diagnosis of PDAC is expected to contribute to the improvement of its prognosis.

## 3. Risk Factors

According to Japanese Clinical Guidelines for PDAC in 2016 [8], risk factors for PDAC include a family history of pancreatic carcinoma, hereditary pancreatic carcinoma syndrome, intraductal papillary mucinous neoplasms (IPMNs), chronic pancreatitis, diabetes, alcohol consumption, and smoking. Patients with several risk factors should be further examined to detect the pancreatic carcinoma.

### 3.1. Family History or Hereditary Pancreatic Carcinoma Syndrome

Familial pancreatic carcinoma is defined as “PDAC arising in a family with two or more first-degree relatives (parents, siblings, and/or children) with PDAC.” Approximately 3–10% of all patients with pancreatic carcinoma have a family history of cancer [9]. Breast and ovarian cancer, Peutz–Jegher, and familial atypical multiple mole melanoma syndromes are defined as hereditary pancreatic carcinoma syndromes. Evaluating patients at high risk for PDAC is important to elucidate the mechanism of carcinogenesis and genetics [10,11]. Canto et al. performed diagnostic imaging on patients at a high risk for PDAC from either hereditary pancreatic carcinoma syndrome or a family history of PDAC, and demonstrated the usefulness of endoscopic ultrasonography (EUS) for identifying pancreatic lesions [12]. Canto et al. recently reported that long-term pancreatic cancer screening programs could detect resectable PDAC and improve the prognosis of high-risk individuals [13]. Harinck et al. revealed that magnetic resonance imaging (MRI) and EUS played complementary roles in the detection of pancreatic lesions in patients at high risk for PDAC, including hereditary pancreatic carcinoma syndrome [14]. These reports demonstrated the importance of systematic screening for patients at a high risk for PDAC.

### 3.2. IPMN

Patients with IPMN are at risk not only for intraductal papillary mucinous carcinoma but also for PDAC concomitant with IPMN [15]. The reported incidence rate of PDAC concomitant with IPMN is 2.0–11.2% [15]. Regular follow-up of IPMN cases is extremely important for the early detection of PDAC [16].

### 3.3. Chronic Pancreatitis

Results of a multicenter, retrospective study conducted in Japan indicated that the standard morbidity ratio of chronic pancreatitis for PDAC is high (11.8%), making it an important risk factor for PDAC [17]. However, screening methods for chronic pancreatitis for the early detection of PDAC have not yet been established. Revised diagnostic criteria for chronic pancreatitis from 2009 [18] introduced the recent concept of early chronic pancreatitis. Early chronic pancreatitis was defined by the Japanese Clinical Diagnostic Criteria for Chronic Pancreatitis, 2009. Early chronic pancreatitis not only satisfies the conditions for definitive or probable diagnoses in diagnostic criteria but also satisfies two or more of the following conditions: (i) repetitive episodes of upper abdominal pain, (ii) abnormal pancreatic enzyme values in the blood or urine, (iii) pancreatic exocrine dysfunction, and (iv) history of persistently consuming ≥80 g of alcohol per day. In addition, a patient should also exhibit specific EUS findings of Rosemont classification [19] and imaging findings of pancreatic duct of Cambridge classification [20]. According to this concept, pancreatitis should be diagnosed and treated at the early stages to prevent progression to chronic pancreatitis. The diagnosis of early chronic pancreatitis will provide opportunities to detect early-stage PDAC associated with chronic pancreatitis [21].

### 3.4. Tobacco and Alcohol

Lifestyle-related factors, particularly tobacco smoking and alcohol consumption, affect the risk of PDAC. Based on the meta-analysis on smoking as a risk factor for PDAC [22], the pooled relative risk (RR) of current and former smokers was 1.66 (95% confidence interval (CI), 1.38–1.98) and 1.40 (95% CI, 1.16–1.67) when compared with never smokers, respectively. On the contrary, a meta-analysis on alcohol consumption as a risk factor for PDAC revealed that the pooled RR of heavy drinkers and former smokers was 1.66 (95% CI, 1.38–1.98) and 1.40 (95% CI, 1.16–1.67) when compared with never drinkers, respectively [23]. However, the mechanisms underlying the effects of tobacco and alcohol have not been clarified yet. Information regarding tobacco smoking and alcohol consumption being risk factors for PDAC should be disseminated to the public. 

### 3.5. Diabetes Mellitus

Diabetes mellitus (DM) is considered one of the risk factors for PDAC. Many studies have been conducted to reveal the relationship between and the mechanism of DM and PDAC [24]. A meta-analysis revealed the relationship between DM and increased risks of PDAC in both males and females [25]. A recent study revealed that new onset diabetes can potentially indicate early stage PDAC [26]. Further studies for identifying the pathogenic mechanisms of the relationship between DM and PDAC are warranted. 

## 4. Attempts to Achieve Early Diagnosis of Pancreatic Carcinoma

The identification of various risk factors and advances in diagnostic imaging have increased the number of pancreatic carcinoma cases reported in early stages [27,28]. Recently, the reported number of stage 0 cases with high-grade pancreatic intraepithelial neoplasia (PanIN) has also been increasing. The factors that need to be considered in the detection of stage 0 cases of PDAC are described below:

### 4.1. PanIN

In 2001, PanIN was defined as the precursor lesion of PDAC. PanIN manifests as consequential development of low to high-grade lesions. Recently, these lesions have been confirmed to have several alterations in *KRAS*, *TP53*, *p16,* and *SMAD4*. Patients with high-grade PanIN, as shown in Figure 3, are expected to have long-term survival [29]. Such patients survive without recurrence for 6 years postoperatively. Egawa et al. [7] and Kanno et al. [30] revealed the 5-year survival rate for stage 0 PDAC as 85.8% and 94.7%, respectively. Thus, accumulating cases of high-grade PanIN is crucial to understanding its causation, which would dictate appropriate treatment and improve the prognosis of pancreatic carcinoma.

### 4.2. microRNA and Cancer-Derived Exosomes

Recently, microRNAs (miRNAs) have gained attention as molecules involved in cancer progression. MiRNA are small, approximately 19–25 nucleotides long non-coding RNAs that post-transcriptionally regulate gene expression [31]. Additionally, several studies have reported that exosomes contribute to tumor cell proliferation by supporting cancer cells with anti-apoptotic protein. Some studies have tried to use these molecules as diagnostic tools [32]. Interestingly, a combined evaluation of serum exosomes expressing proteins and miRNA markers revealed that PDAC patients present a distinct pattern of exosomes and miRNA markers, thus providing a novel diagnostic strategy [33]. In the near future, it will become possible to use several molecules as miRNA or exosomes in serum or pancreatic juice in liquid biopsy.

### 4.3. Hematological and Biochemical Tests and Tumor Markers

Hematological and biochemical tests are usually nonspecific. However, abnormal findings sometimes help diagnose PDAC. The high serum levels of pancreatic enzymes often lead to PDAC diagnosis [34]. The purpose of the high serum levels of pancreatic enzymes is to obstruct the pancreatic duct. Further imaging examinations should be performed when cases with high serum levels of pancreatic enzymes are found. 

Tumor markers are neither tumor-specific nor pancreatic cancer-specific. The glycoprotein carbohydrate antigen 19-9 (CA19-9) is one of the important tumor markers for PDAC [35]. Elevated CA19-9 serum levels are useful as poor predictors of PDAC. However, CA19-9 is not suitable as a screening tool for detecting early-stage PDAC in asymptomatic patients. The combined use of other tumor markers, such as carcinoempryonic antigen (CEA), DUPAN 2, or Span 1, is very important for diagnosing PDAC. 

### 4.4. Importance of Endoscopic Retrograde Cholangiopancreatography (ERCP) for Early Diagnosis of Pancreatic Carcinoma

The use of endoscopic retrograde cholangiopancreatography (ERCP) in the diagnosis of pancreatic carcinoma has conventionally been avoided owing to its low diagnostic ability [36] and associated risks for post-ERCP pancreatitis [37,38]. Thus, EUS, which provides clear images without the risk of post-ERCP pancreatitis, is an essential modality for diagnosing PDAC. In particular, EUS-fine needle aspiration (EUS-FNA) has enabled the histopathological diagnosis of PDAC, dramatically changing the diagnostic algorithm for PDAC. However, EUS-FNA can barely detect cases of high-grade PanIN without a pancreatic mass. Moreover, ERCP is an important diagnostic modality for early-stage PDAC, which involves aggressive performance of pancreatic juice cytology using endoscopic nasopancreatic drainage (ENPD) [27]. Based on Japanese clinical data for PDAC [8], the Japan Pancreas Society established the Japanese Clinical Guideline for PDAC. In this guideline, ERCP was emphasized for detecting early-stage PDAC in Clinical Question-D7 and in the diagnostic algorithm, as shown in Figure 4. Issues with selecting cases demonstrating changes in the main pancreatic duct and performing ERCP and ENPD by reducing the risk of post-ERCP pancreatitis should be addressed in the future.

### 4.5. Cooperation of Local Clinics and Regional Hospitals

The selection of cases requiring further examination is very important in the diagnosis of early-stage PDAC. The Pancreatic Carcinoma Early Diagnosis Project, also known as the Onomichi Project, was established in 2007 and is based on the cooperation of the JA Onomichi General Hospital and local clinics. Many cases of early-stage PDAC have been diagnosed via this project [39]. The doctors of the JA Onomichi General Hospital have delivered lectures on the risk factors of PDAC, ultrasonography (US) of the pancreas, and importance of US screening and EUS diagnostic imaging to disseminate information on the clinical characteristics of PDAC. Patients with abnormal US findings examined by local doctors were referred to the JA Onomichi General Hospital for further examination. A model case, such as the Onomichi Project, should be employed in other areas for the detection of early-stage PDAC (Figure 5).

### 4.6. Japan Study Group on the Early Detection of Pancreatic Cancer (JEDPAC)

The Japan Study Group on the Early Detection of Pancreatic Cancer (JEDPAC) was established in 2014 to clarify the clinicopathological features of pancreatic carcinoma diagnosed in the early stages. The imaging and pathological characteristics of early-stage PDAC have been gradually investigated by this study group. This study will elucidate the clinical and genetic findings and contribute to the improvement of the prognosis of PDAC.

## 5. Current Status of Pancreatic Carcinoma Diagnosed in the Early Stages

The clinical features of early-stage PDAC need to be clarified to diagnose many cases of PDAC in its early stages. JEDPAC investigated the clinical findings of 51 cases of stage 0 pancreatic carcinoma and 149 cases of stage 1 pancreatic carcinoma [30]. The features of PDAC diagnosed in the early stages have been gradually clarified based on the results of this survey, as shown in Table 2. PDAC diagnosed in the early stages had several risk factors, including diabetes in 64 cases (32%), smoking in 62 (31%), and IPMN in 52 (26%), as shown in Table 3. Early-stage PDAC was diagnosed via further investigation of these cases, which revealed the presence of symptoms in 50 cases (25.0%), abnormalities during examination or follow-up for other diseases in 103 (51.5%), and abnormalities during medical health checkup in 34 (17%), as shown in Figure 2. Various diagnostic imaging modalities were used for further examining of stage 0 and 1 PDAC cases. CT, MRI, and EUS were performed for >80% of all cases. However, most abnormal findings detected via these diagnostic imaging modalities were indirect findings, such as main pancreatic duct dilatation or retention cysts, without direct findings of PDAC, as shown in Table 4 and Figure 6. Thus, the detection of indirect findings on diagnostic imaging in asymptomatic cases and further examination on such cases is important for the early diagnosis of PDAC. Further examinations using EUS, ERCP, and EUS-FNA were performed in 173 (86.5%), 141 (70.5%), and 69 cases (34%), respectively. These results indicated the importance of ERCP in the early diagnosis of pancreatic carcinoma. The diagnostic strategies for the early diagnosis of PDAC should be constructed based on these results.

## 6. Conclusions

Recent investigations have revealed several clinical characteristics of PDAC diagnosed in its early stages. Cooperation between local clinics and regional hospitals is crucial to detect PDAC in its early stages. To acquire this cooperation, one should provide the clinical findings of early-stage PDAC to local clinics so that they can refer cases with abnormal findings to regional hospitals. Data on the risk factors or imaging findings of early-stage PDAC are very important. Especially, ERCP is an important diagnostic modality for early-stage PDAC to perform pancreatic juice cytology using ENPD after detecting abnormal findings. However, because the potential for the diagnosis of early-stage PDAC is limited, new approaches are warranted.

## Figures and Tables

**Figure 1 diagnostics-09-00018-f001:**
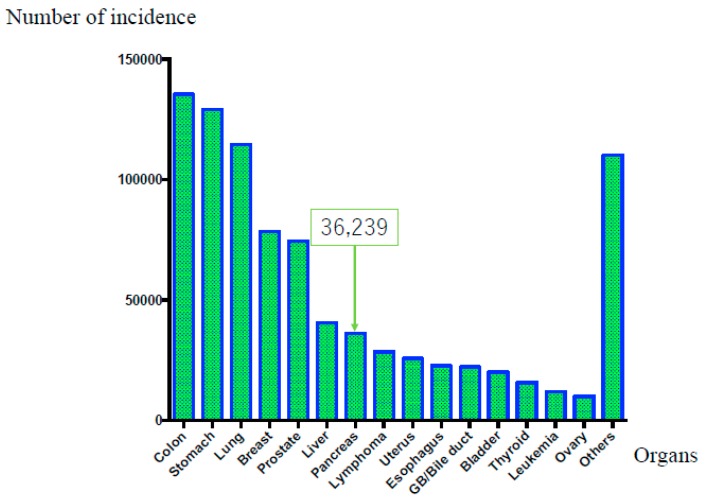
Overall cancer incidence rate according to affected organ (in Japan). The overall cancer incidence in 2014 was as follows: (1) colorectal cancer, 135,434 cases; (2) gastric cancer, 129,239 cases; (3) lung cancer, 114,550 cases; (4) breast cancer, 78,529 cases; (5) prostate cancer, 74,459 cases; (6) liver cancer, 40,666 cases; and (7) pancreatic carcinoma, 36,239 cases. Thus, pancreatic ductal adenocarcinoma occupies seventh place.

**Figure 2 diagnostics-09-00018-f002:**
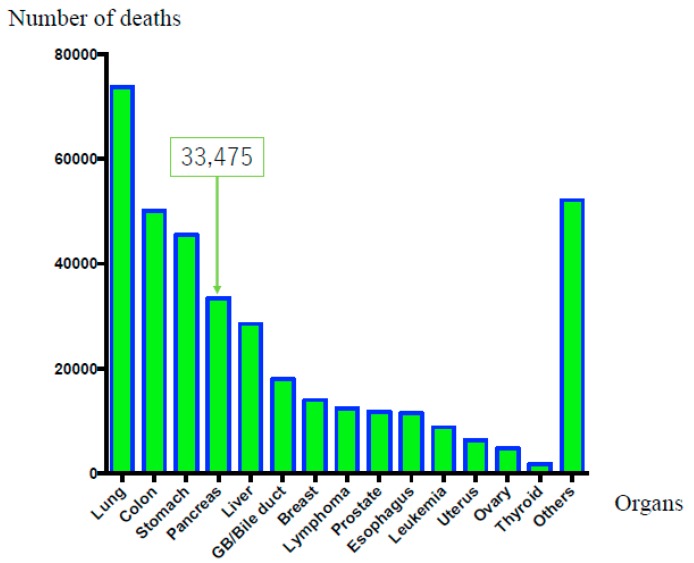
Mortality rate in according to the affected organ (in Japan). The overall numbers of cancer-related deaths in 2016 were as follows: (1) lung cancer, 73,838 cases; (2) colorectal cancer, 50,099 cases; (3) gastric cancer, 45,531 cases; and (4) pancreatic carcinoma, 33,475 cases (men: 17,060 cases, women: 16,415 cases).

**Figure 3 diagnostics-09-00018-f003:**
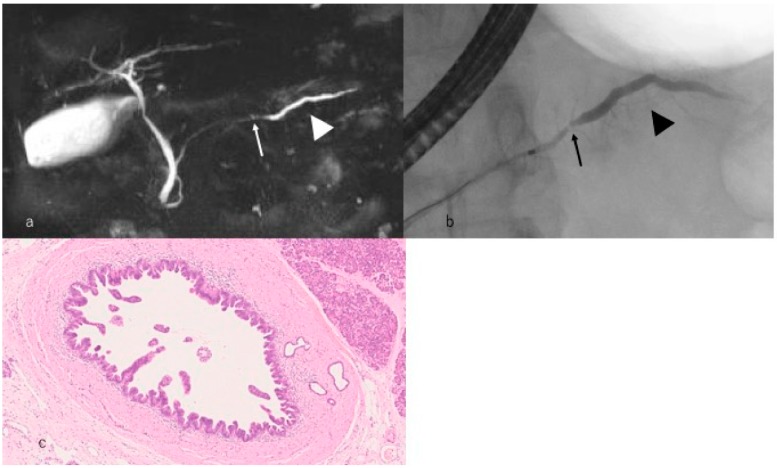
A case with high-grade pancreatic intraepithelial neoplasia. (**a**) Magnetic resonance cholangiopancreatography reveals that the main pancreatic duct is narrowed (arrow) in the pancreatic body and the caudal side pancreatic duct is mildly dilated (arrowhead). (**b**) Endoscopic retrograde cholangiopancreatography reveals that the main pancreatic duct is locally narrowed (arrow) in the pancreatic body and the caudal side pancreatic duct is mildly dilated (arrowhead). (**c**) Histopathological imaging reveals intraepithelial cancer in the main pancreatic duct.

**Figure 4 diagnostics-09-00018-f004:**
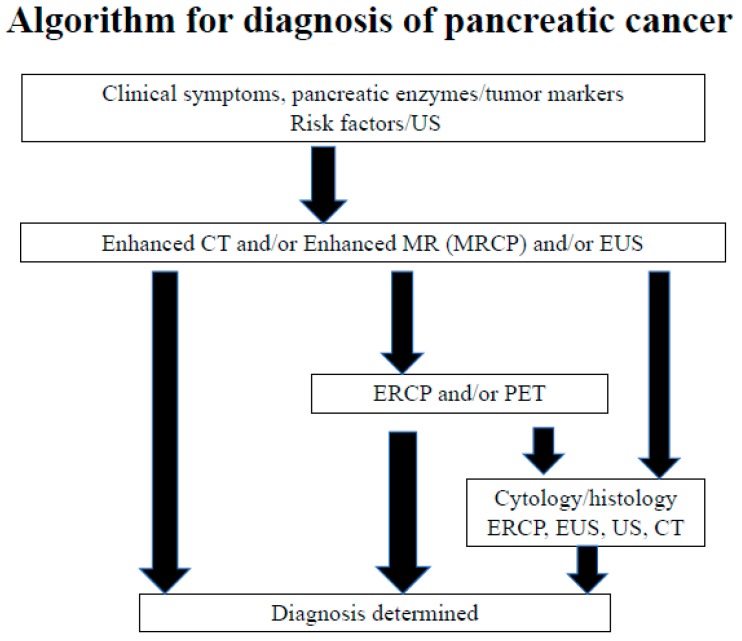
Algorithm of pancreatic cancer diagnosis (adopted from Reference [7]).

**Figure 5 diagnostics-09-00018-f005:**
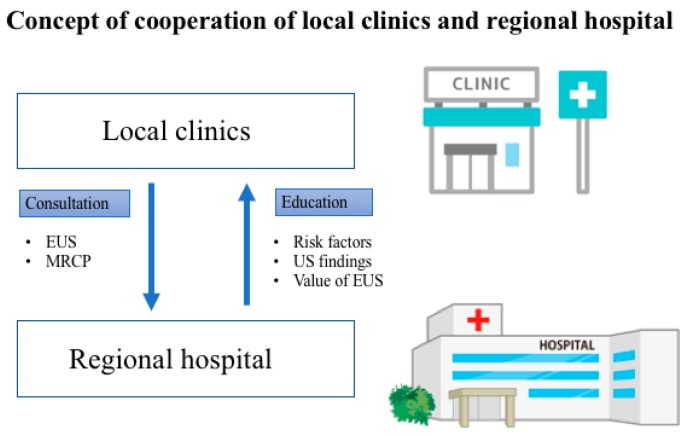
Concept of cooperation of local clinics and regional hospitals. Networks between local clinics and regional hospitals should be established to detect early-stage pancreatic cancer. EUS, endoscopic ultrasonography; MRCP, magnetic resonance cholangiopancreatography; US, ultrasonography.

**Figure 6 diagnostics-09-00018-f006:**
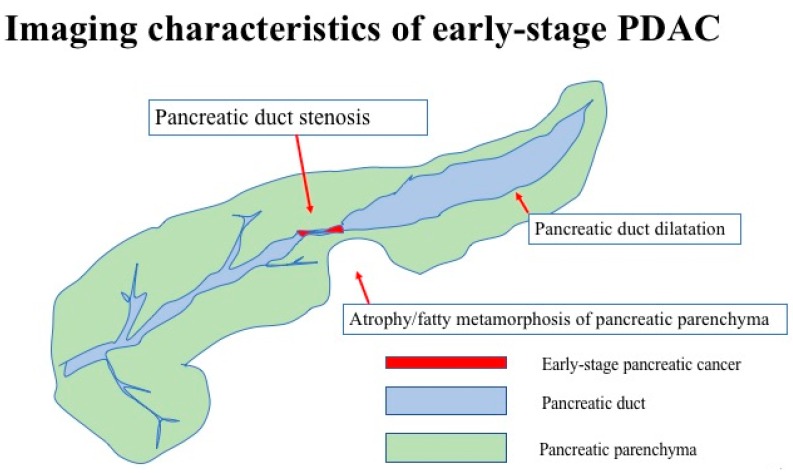
Imaging characteristics of early-stage PDAC.

**Table 1 diagnostics-09-00018-t001:** Comparison of incidence and mortality rate of pancreatic cancer in the world and in Japan.

Incidence and Mortality Rate of Pancreatic Cancer	Worldwide	Japan
Incidence rate (male/female) (/100,000)	4.9/3.6	10.6/6.7
Mortality rate(male/female) (/100,000)	4.8/3.4	8.4/6.1

**Table 2 diagnostics-09-00018-t002:** Clinical background of patients (quoted from Reference [23]).

Sex (Male/Female)	111/89
Age, mean ± SD (range), years	68.8 ± 9.5 (38–88)
Stage (0/1), number of cases	51/149
Observation period, mean (range), days	1240.8 (66–3635)
Location, head/body/tail, number of cases (%)	86 (43.0)/103 (51.5)/11 (5.5)

**Table 3 diagnostics-09-00018-t003:** Risk factors observed in patients (quoted from Reference [23]).

Risk Factors (Including Overlapping Cases)	Number of Cases (%)
Diabetes	64 (32.0)
Smoking	62 (31.0)
Intraductal papillary mucinous neoplasm	52 (26.0)
Heavy Alcohol consumption	30 (15.0)
Chronic pancreatitis	26 (13.0)
Obesity	13 (6.5)
Family history of pancreatic carcinoma	9 (4.5)

**Table 4 diagnostics-09-00018-t004:** Diagnostic imaging findings (quoted from Reference [23]).

Diagnostic Imaging Modalities and Findings		Number of Patients (%)
**Abdominal ultrasound**		135/200 (67.5)
Findings (some overlapping cases)	Pancreatic duct dilatation	101/135 (74.8)
	Pancreatic duct stenosis	27/135 (20.0)
	Pancreatic tumors	71/135 (52.6)
**Computed tomography**		196/200 (98.0)
Findings (some overlapping cases)	Pancreatic duct dilatation	156/196 (79.6)
	Pancreatic tumors	101/196 (51.5)
	Atrophy/fatty metamorphosis of pancreatic parenchyma	82/196 (41.8)
**Magnetic resonance imaging**		173/200 (86.5)
Findings (some overlapping cases)	Pancreatic duct dilatation	143/173 (82.7)
	Pancreatic tumors	78/1733 (45.1)
**Endoscopic ultrasonography**		173/200 (86.5)
Findings (some overlapping cases)	Pancreatic duct dilatation	153/173 (88.4)
	Pancreatic duct stenosis	98/173 (56.6)
	Pancreatic tumors	132/173 (76.3)

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
