# Peer review of "Advances in Early Detection of Pancreatic Cancer"

_diagnostics, 2019, doi:10.3390/diagnostics9010018_

Reviewer 1 Report

The authors of this manuscript investigate a very important problem in medicine: Advances in early diagnosis of PDAC. I identified some minor and some major issues in the paper that need to be addressed, which are listed below:

1. Lines 29-30:

The following sentence needs to be clarified: “However, the number of deaths caused by PDAC is gradually increasing, affecting approximately 44,330 individuals.” The authors need to point out that these are average annual deaths, if that is the case. Are these numbers in the USA or worldwide? In the previous sentence, the authors were describing cancer statistics in the USA. In the following sentence, the authors indicated that this high incidence of PDAC makes it to be the fourth leading cause of death worldwide. This issue also needs to be clarified.

2. Lines 30-32:

The authors need to clarify that PDAC is the fourth leading cause of death by cancer worldwide, as opposed to the fourth leading cause of death worldwide (from all causes).

3. Lines 33-37:

The authors need to explicitly state that the overall statistics for cancer incidence in 2014 are from Japan. A reference to Fig. 1 with the graph displaying these statistical numbers should be included in the text.

4. lines 44-46:

The authors need to explicitly state that the overall number of cancer-related deaths in 2016 are from Japan. A reference to Fig. 2 with the graph displaying these statistical numbers should be included in the text.

5. In Line 93:

The authors need to display the number (11.8) as a percentage (11.8%)?

6. In Line 154:

If possible, the authors need to clarify if the diabetes was the cause or the symptom of PDAC in the patients with diabetes. That could be done if the number of patients that had diabetes many years prior to being diagnosed with PDAC is identified out of the population of 64 patients studied.

7. In Lines 173-177:

The primary conclusions reached by the authors do not contribute to the advancement in the field. These conclusions are:

“Furthermore, cooperation between local clinics and regional hospitals is crucial to detect PDAC in its early stages. New approaches for the early-stage diagnosis of PDAC are warranted.”

In the first sentence, the authors indicated that cooperation is crucial. However, no specific details on this cooperation was provided. In the second sentence, no potential solution that could be further investigated was identified by the authors when they pointed out the need for new approaches for early-state diagnosis.

8. The authors also did not mention any progress related to the identification and use of molecular biomarkers of PDAC such as CA19-9 that could be used in the diagnosis or in the prognosis of pancreatic cancer in lab tests. This topic has to be covered in any paper that is focusing on the advances of early detection of PDAC, even if lab tests for the diagnosis and prognosis of PDAC have not yet achieved the level of success than equivalent lab tests for other types of cancer.

Author Response

1. We apologize for the errors. We have revised the descriptions according to your suggestion on page 3, line 13.

2. We have revised the descriptions according to the suggestion on page 3, line 13.

3. As suggested, we have added the overall statistics for cancer incidence in 2014 on page 4, lines 1–5.

4. As suggested, we have added the overall statistics for cancer-related deaths in 2016 on page 4, lines 10–14.

5. We have added the “%” on page 3, line 13.

6. As suggested, we have added an explanation on the relationship between DM and PDAC on page 8, lines 4–9. We have also added reference #23 and #24. However, we could not revise the description of the 64-patients population studied as we could not obtain data in detail due to these data haven been collected via questionnaires.

7. As suggested, cooperation between local clinics and regional hospitals is crucial. To acquire this cooperation, we should provide the clinical findings of early-stage PDAC to local clinics so that they can refer such cases with abnormal findings to regional hospitals. Data on the risk factors or imaging findings of early-stage PDAC are very important. However, it is difficult to diagnose early-stage PDAC because findings are limited; hence, new method are needed. The relevant sentences have been added to page 13, lines 3–7.

8. As suggested, we have added the description on hematological and biochemical tests and tumor markers on page 9, line 17–page 10, line 7.

Reviewer 2 Report

The article provides a nice review of pancreatic cancer early detection efforts and outcomes. Overall, it seems a bit light on details and has some confusing usage of statistics. Some notes below:

1. In the Epidemiology section, the authors mention that the number of patients diagnosed with lung/colon cancer in the United States each year is going down. This is incorrect. The numbers are going up slightly year-over-year. Perhaps they were referring to the incidence rate rather than absolute number of diagnoses.

2. Also, it's unclear whether the authors switch from U.S.-based stats to worldwide. It is incorrect to say that pancreatic cancer is the fourth leading cause of death worldwide. It is the seventh leading cause of cancer related-death worldwide and third in the United States.

3. The Japan-based stats are also unclear, switching between 2016 and 2014. 

4-a.The 11.6 percent five-year survival rate in Japan seems unexpectedly high. Reference #4 does not seem to include a full-text option, so I was unable to verify the source of that stat.

4-b. Similarly, and citing the same reference, stating that stage 0, Ia and Ib five-year survival rates are 85.8, 68.7 and 59.7 percent, respectively, seems exceptionally high.

5. In figure 2, the arrow appears to be pointing to liver rather than pancreas.

6. In section 3.1, the authors could consider referencing a more recent study from Canto, et al, showing the value of EUS-based screening for high-risk individuals: https://www.ncbi.nlm.nih.gov/pubmed/29803839.

7. In section 3.3, it's unclear how a clinician may differentiate "early chronic pancreatitis" from an acute case.

8. In the intro to section 4, the authors say that identification of various risk factors and advances in diagnostic imaging have increased the number of PDAC cases reported in the early stage - it's important to clarify that this is occurring within a research/screening setting and not in the general population (yet).

9. In section 4.1, it should be clarified whether the six years' survival post-operatively for patients diagnosed with high-grade PanIN is an average. 

10. It is unclear why section 4.2 is touting ERCP whereby earlier, the authors mention successful screening efforts utilizing EUS or MRI. Also not clear how/whether ENPD can mitigate ERCP-induced pancreatitis.

11. In the intro to section 5, the authors mention patients diagnosed with early-stage pancreatic cancer who also have diabetes. It's not specified whether this is longstanding diabetes, which is a mild risk factor for PDAC, or a recent onset of diabetes, which can be considered a symptom (see https://www.ncbi.nlm.nih.gov/pmc/articles/PMC5495015/ and others).

Author Response

1. We have revised the descriptions in line with your suggestions on page 3, line 13.

2. We have revised the descriptions in line with your suggestions on page 3, line 13. Furthermore, to compare the Japanese and global data, we have included the GLOBACON, 2012 data on pancreatic cancer in Table 1 and have revised the sentences on page 3, lines 13–17.

3. The Japanese cancer epidemiology data have a time lag between the incident and mortality rates. The mortality rate in 2016 and the incident rate in 2014 are the latest data of the Japanese cancer statistics. Please refer to the web page of the Japanese cancer center (https://ganjoho.jp/reg_stat/statistics/qa_words/statistics_qa.html).

4-a. As suggested, we re-examined reference #4(new reference number #6). Based on Figures 5 and 6 of reference #4, we have revised the data and description on page 4, lines 12–14.

4-b. As suggested, these data are favorable. However, the proportion of early-stage cases was low [stage 0: 411/23582 (1.7%), stage Ia:969/23582 (4.1%), stage Ib :1484/23582 (6.3 %)] in Figure 14 of reference #4 (new reference number #6). These data show the difficulty to diagnose early-stage PDAC. We have tried to increase the number of cases with early-stage PDAC (page 5, lines 1–8).

5. We have revised the figure as indicated.

6. As suggested, we have added descriptions regarding the new study of Canto et al. on page 6, lines 4–6.

7. Early chronic pancreatitis was defined by the Japanese Clinical Diagnostic Criteria for Chronic Pancreatitis 2009. Early chronic pancreatitis not satisfy the condition for definitive or probable diagnosis in diagnostic criteria but also satisfies two or more of the following conditions: (i) repetitive episodes of upper abdominal pain, (ii) abnormal pancreatic enzyme values in the blood or urine, (iii) pancreatic excrine dysfunction, and (iv) history of persistently consuming 80 g of alcohol per day. In addition, a patient should also exhibit specific EUS findings. The explanation on early chronic pancreatitis has been added to page 7, lines 1–9.

8. Thank you for your insightful comment. However, the findings of the risk for PDAC in cases with risk factors are limited. Canto et al. reported that long-term pancreatic cancer screening programs could detect the resectable PDAC and improve the prognosis of high-risk individuals with hereditary pancreatic carcinoma syndrome or family history of PDAC. The risk to for PDAC to occur concomitant with IPMN is reportedly 2.0%–11.2%. We need to prospectively clarify this risk for PDAC occurrence in the population with each risk factor. We have added the sentences of Canto’s report on page 6, lines 4–6.

9. Egawa et al. (new ref #6) and Kanno et al (new ref #28) revealed the 5-year survival rate for stage 0 PDAC (ref #4) as 85.8% and 94.7%, respectively. We have added this description on page 8, line 23– page 9, line 2.

10. EUS-FNA is useful diagnostic modality for the histological diagnosis of PDAC. However, it is impossible to diagnose PDAC with only localized stenosis or caliber change of pancreatic duct. ERCP is needed for histological diagnosis of these cases. IIboshi et al. revealed the usefulness of cytological diagnosis using ENPD. Based on these data, the clinical guideline for PDAC established by the Japan Pancreas Society recommends pancreatic juice cytology using ENPD for detecting early-stage PDAC in CQ-D7. As mentioned, ERCP has the risk of post-ERCP pancreatitis (PEP). Though the frequency of PEP using ENPD is low, the safety of ENPD for diagnosis of PDAC should be clarified in the future. We have added the relevant descriptions on page10, lines 20–22.

11. As explained to reviewer 1 (6), we have added an explanation on the relationship between DM and PDAC on page 8, lines 4–9 and references #23 and #24, as suggested.

Reviewer 3 Report

Kanno et al have attempted to describe advancements in early detection of pancreatic cancer. 

However, most statements lack details. Early diagnosis of pancreatic cancer is a huge field and there are innumerable reports that are published routinely. Merely, quoting that further approaches are warranted, does not end the job of research community. Put forth your views on recently published approaches, find loop holes and suggest ways of overcoming those.

Also, it is disappointing to see the authors replicating their own published tables (Ref # 23). 

Overall, I believe that a substantial improvement in the manuscript in terms of adding scientific novelty is required.

Author Response

Thank you for helpful and insightful comment. One of associated editors asked us to write a review manuscript on the early diagnosis of PDAC (regarding imaging findings and epidemiology). We have showcased our data with a focus on the imaging findings of JEDPAC (new ref #28). As suggested by the other reviewers, we have added several explanations to overcome the challenges of diagnosing PDAC in the early stages.

Round  2

Reviewer 1 Report

This manuscript reviews recent advances in the early diagnosis of PDAC, which is an important problem in medicine, and present recent statistical results related to PDAC and other types of cancer in Japan and other countries. The revised version of the manuscript addresses the concerns that were raised in the original review of the paper.

On line 128, I suggest that the words “should be spread to the public” be replaced with “should be disseminated to the public”.

Author Response

As suggested, We have revised the descriptions according to your suggestion on line 127.

Reviewer 2 Report

Thank you for your thorough responses to the previous comments and recommendations. My only slight hesitation remains the diabetes section - i think there is ample evidence (e.g., https://www.ncbi.nlm.nih.gov/pubmed/30325864) that new-onset diabetes can also serve as a symptom of pancreatic cancer, making this information important to include within the scope of this review article. 

Author Response

As suggested, we have added the descriptions of new-onset diabetes and pancreatic cancer on line 132-133.

Reviewer 3 Report

1. Line 29 : Globacon has published a more recent statistics in September 2018. Please use that number (https://doi.org/10.3322/caac.21492)     

2. There is very less novelty in this review compared to information already available. however, I would be willing to give a positive response, if the Conclusion is re-framed to include more details rather than general/vague sentences.  The take home message is not concrete.

Author Response

1. As suggested, we have changed the data of recent cancer statistics on line 30-31. However, it is impossible to compare the epidemiological data of pancreatic cancer between Japan and other countries based on GLOBACON 2018. Table 1 using the data of GLOBACON 2012 has not been changed because the data of GLOBACON 2012 reveals some differences between the Japanese and global data.  

2. Thank you for helpful and insightful comment. As suggested, we have added the descriptions of importance of ERCP for detecting early stage PDAC to add the concrete comment on line 239-241.